# Fasciola hepatica excretory-secretory products attenuate demyelination and reduce neuroinflammation in the Cuprizone –induced multiple sclerosis model

Aliakbar Mariki[1], Kristi Anne Kohlmeier[2], Seyed Mohammad Mousavi[3], Alireza Keyhani[4], Majid Fasihi Harandi[3], Mansoureh Sabzalizadeh[1,5], Mohammad Shabani[1]*

**1** Neuroscience Research Center, Institute of Neuropharmacology, Kerman University of Medical Sciences, Kerman, Iran, **2** Department of Drug Design and Pharmacology, Faculty of Health Sciences, University of Copenhagen, Copenhagen, Denmark, **3** Research Center for Hydatid Disease in Iran, School of Medicine, Kerman University of Medical Sciences, Kerman, Iran, **4** Leishmaniasis Research Center, Kerman University of Medical Sciences, Kerman, Iran, **5** Cognitive Neuroscience Research Center, Institute of Neuropharmacology, Kerman University of Medical Sciences, Kerman, Iran

\* shabanimoh@yahoo.com, shabani@kmu.ac.ir

## Abstract

Multiple sclerosis (MS) is characterized by chronic neuroinflammation and progressive demyelination, with current therapies failing to adequately address both processes simultaneously. Helminth-derived excretory–secretory products (ESP) are immunomodulatory molecules that suppress host inflammation and are promising candidates for MS treatment. However, their capacity to promote remyelination remains poorly understood. This study investigated the effects of Fasciola hepatica ESP in the cuprizone-induced demyelination model. Male C57BL/6 mice were divided into four groups: control (CON), control+FES (CON+FES), cuprizone (CUP), and cuprizone+FES (CUP+FES). FES was administered intraperitoneally during the demyelination phase. Motor and cognitive functions were evaluated using open field, rotarod, wire grip, and shuttle box tests. Neuroinflammation was assessed by measuring TNF and IL-1β levels, while myelin-related changes were evaluated by MBP and Olig2 expression (ELISA and qPCR) and Luxol Fast Blue staining. FES treatment significantly reduced pro-inflammatory cytokines, increased MBP and Olig2 levels, improved myelin integrity, and enhanced motor and cognitive performance compared to untreated cuprizone mice, though full recovery to control levels was not achieved. These findings demonstrate that FES attenuates neuroinflammation and is associated with partial improvements in myelin-related outcomes in the cuprizone model, supporting the therapeutic potential of helminth-derived molecules for MS.

**Data availability statement:** All relevant data for this study are publicly available from the OSF repository (https://osf.io/576q4).

**Funding:** The author(s) received no specific funding for this work.

**Competing interests:** The authors have declared that no competing interests exist.

## Introduction

Multiple sclerosis (MS) is a chronic autoimmune disorder that affects the central nervous system (CNS), including the brain and spinal cord. In MS, the immune system mistakenly attacks the protective myelin sheath surrounding nerve fibers, leading to inflammation and subsequent neuronal damage [1]. This process disrupts communication between the brain and other parts of the body, resulting in clinical manifestations such as fatigue, muscle weakness, numbness, visual disturbances, impaired balance, and cognitive dysfunction [2]. The severity of MS varies, with the most common form being relapsing-remitting MS (RRMS) [3], characterized by episodes of symptom exacerbation followed by periods of partial recovery. Other types include secondary progressive MS (SPMS) [4] and primary progressive MS (PPMS) [5]. The precise cause remains unclear, but a combination of genetic and environmental factors is thought to play a role in the onset of the disease [6]. In MS, factors such as chronic inflammation, oxidative stress, mitochondrial dysfunction, and axonal injury further exacerbate remyelination failure [7]. Persistent activation of microglia within MS lesions has also been shown to create an inhibitory microenvironment for oligodendrocyte precursor cells (OPCs), thereby preventing their proper differentiation and efficient remyelination [8]. Currently, there is no definitive cure for MS. However, treatments such as immunomodulatory therapies, physiotherapy, and lifestyle modifications may help control symptoms and slow disease progression. Early diagnosis and timely intervention play a critical role in improving patients' quality of life. While current therapies primarily focus on immune modulation, recent advances in neuroprotection and pro-remyelination strategies, such as targeting growth factors and enhancing OPC recruitment, offer promising avenues for repairing myelin damage in MS [9]. A promising area in neuroimmunology research lies in the potential of helminth-derived molecules to modulate immune responses and enhance neuroprotection [10]. Notably, *Fasciola hepatica*, a trematode parasite with remarkable immunomodulatory capacity, has received increasing attention [11,12]. This parasite secretes a diverse array of excretory-secretory products (ESPs), including proteases such as cathepsin L, antioxidant enzymes like peroxiredoxins and glutathione S-transferases, lipid-binding proteins, Kunitz-type protease inhibitors, and the helminth defense molecule FhHDM-1, as well as extracellular vesicles enriched with microRNAs [13]. These molecules employ multiple mechanisms to modulate immunity, including downregulation of pro-inflammatory pathways (e.g., NF-κB signaling), inhibition of inflammasome activation, suppression of autoreactive T and B cell responses, promotion of regulatory T cell expansion, Th2 polarization, and alternative activation of macrophages toward an anti-inflammatory M2 phenotype [14,15]. In chronic inflammatory conditions like MS, where microglial activation and oxidative stress hinder OPC survival and differentiation, the anti-inflammatory and antioxidant properties of *F. hepatica* ESPs could help create a more permissive environment for remyelination. By reducing pro-inflammatory cytokines, boosting anti-inflammatory mediators, minimizing oxidative damage, and promoting a reparative glial phenotype, these secretions could restore conditions for OPC recruitment, maturation, and myelin repair [16,17]. With no clear evidence to date that helminth-derived products

directly promote repair in MS, we turned to the cuprizone model of demyelination to probe the capacity of *Fasciola hepatica* ESPs to restore myelin and rescue cognitive and motor function. Cuprizone administration induces reproducible oligodendrocyte loss and CNS demyelination and is widely used as a toxin-induced (non-immune) model to study mechanisms of demyelination and remyelination relevant to MS [18]. This model involves mitochondrial dysfunction and severe oxidative stress in oligodendrocytes due to copper chelation, leading to selective apoptosis independent of adaptive immune responses [19]. The hippocampus and cerebellum were chosen as primary regions of interest due to their marked vulnerability to cuprizone-induced demyelination and their well-established involvement in MS. Demyelination within the hippocampus has been closely linked to cognitive deficits in MS, while pathological changes in the cerebellum predominantly contribute to motor impairment and ataxia. To further strengthen our observations, we analyzed inflammatory mediators and myelination markers. Our findings provide insights into the therapeutic potential of helminth-derived products for supporting myelin-related recovery in MS.

## Materials and methods

### Animal

Male C57BL/6 mice (n=28, 6 weeks old, weighing 20–25 g) were used in this study to minimize variability associated with hormonal fluctuations, which may influence neuroinflammatory and remyelination processes, and were purchased from the Animal Breeding Facility of Kerman University of Medical Sciences, Iran, and maintained under standard laboratory conditions (12 h light/dark cycle, temperature 22±2 °C, and relative humidity 50–60%) with ad libitum access to food and water. The animals were randomly divided into four groups (n=7 per group): 1) Control group (CON), healthy mice with no treatment; 2) Control+Fasciola excretory–secretory product group (CON+FES), healthy mice receiving only FES products; 3) Cuprizone group (CUP), mice fed a cuprizone diet to induce demyelination; 4) Cuprizone+Fasciola excretory–secretory product group (CUP+FES), mice fed a cuprizone diet along with *Fasciola hepatica* excretory–secretory (FES) products. Humane endpoints were predefined prior to the start of the study, and animals were monitored daily for signs of severe distress, including marked weight loss (>20%), severe motor impairment, inability to access food or water, or abnormal posture. None of the animals met these humane endpoint criteria during the course of the experiment; therefore, early euthanasia was not required. If any animal had reached the predefined humane endpoint criteria, euthanasia would have been performed immediately. All animals were maintained for the full experimental duration (30 days). A total of 28 mice were used in the study; all 28 animals were euthanized at the planned experimental endpoint, and no unanticipated deaths occurred. Animals were euthanized by carbon dioxide ($CO_2$) inhalation, followed by cervical dislocation to ensure death prior to tissue collection. All experimental procedures were approved by the Ethics Committee of Kerman University of Medical Sciences (Approval No. IR.KMU.AEC.1404.016) and were performed in accordance with institutional ethical guidelines.

### Induction of demyelination in mice using the cuprizone model

Demyelination was induced by feeding mice a cuprizone (bis-cyclohexanone oxaldihydrazone; Sigma-Aldrich, USA)-containing diet mixed into standard chow. Animals received 0.7% (w/w) cuprizone during the first week, followed by 0.2% (w/w) cuprizone for the next three weeks (weeks 2–4). This two-phase regimen was chosen to rapidly initiate oligodendrocyte apoptosis and demyelination in the hippocampus and cerebellum with the higher loading dose, while the subsequent lower maintenance dose sustained the pathology with a reduced risk of systemic toxicity and mortality. The mice were monitored daily, and motor disabilities were assessed weekly using a 0–5 scoring system (0=no symptoms, 1=partial tail weakness, 2=complete tail paralysis, 3=mild hind limb weakness, 4=complete hind limb paralysis, 5=moribund or death) [20].

### Preparation of *Fasciola hepatica* Excretory-Secretory Products) ESPs(

Livers of sheep infected with *Fasciola hepatica* were collected from a slaughterhouse in Kerman, and more than 60 adult flukes were isolated. The parasites were thoroughly washed several times with sterile physiological saline to remove

residual tissues and blood. Subsequently, 5–10 flukes were transferred into each tube containing RPMI-1640 medium supplemented with streptomycin or placed in an anaerobic chamber and incubated at 37 °C for 24 h to allow secretion of proteins. The supernatant was then collected and centrifuged at 4 °C for 1 hour at 4000 rpm. To minimize bacterial contamination and endotoxin levels, the final ES supernatant was filtered through a 0.22 µm sterile syringe filter before protein quantification and use. The protein concentration of the ES antigens was determined using the Bradford method. Briefly, bovine serum albumin (BSA) standards with different concentrations were prepared and added to the wells of a microplate along with the samples. Bradford reagent was added to each well, and after 5 min of incubation at room temperature, absorbance was measured at 595 nm using a microplate reader. The protein concentration of the samples was calculated based on the standard curve, and the final concentration was obtained by multiplying the measured value by the dilution factor [21].

### Drug treatment and administration

*Fasciola hepatica* excretory-secretory products (FES) were administered intraperitoneally (i.p.) at a dose of 10 µg FES in 100 µL of sterile PBS (to maintain physiological pH and preserve protein stability) per injection [22]. The regimen consisted of six injections given on alternate days, starting on day 14 of the cuprizone diet and continuing on days 14, 16, 18, 20, 22, and 24. Mice in the CUP+FES and CON+FES groups received this FES treatment protocol, while mice in the CUP and CON groups were administered 100 µL of sterile PBS i.p. on the same schedule to match handling and injection stress across all groups. All injections were performed using aseptic technique, and mice were monitored for acute adverse reactions after each dose. The dose and intraperitoneal route were selected based on previous studies using *Fasciola hepatica*-derived excretory-secretory components [23].

### Behavioral assessments

**Open field test.** The open field test was performed to assess locomotor activity and anxiety-like behavior in all experimental groups of mice (CUP, CUP+FES, CON, and CON+FES) subjected to cuprizone-induced demyelination and/or FES treatment. The apparatus consisted of a Plexiglas arena (90 × 90 × 45 cm) with the floor divided into 16 equal squares, distinguishing central and peripheral zones. At the beginning of each trial, mice were placed in the central zone and allowed to explore freely for 5 minutes. Parameters such as movement velocity and overall mobility were recorded. After each session, the arena was cleaned with 70% ethanol to remove odor cues [24].

**Rotarod test.** The rotarod test was performed to evaluate motor coordination and muscle strength in all experimental groups of mice (CUP, CUP+FES, CON, and CON+FES) subjected to cuprizone-induced demyelination and/or FES treatment. Mice were habituated to the rotating rod during a training session one day before the main test. On the test day, each mouse underwent three trials on the rod, which accelerated from 10 to 60 RPM. A 30-minute rest period was provided between trials to minimize fatigue. The latency to fall from the rod was recorded for each mouse using a stopwatch [25].

**Wire grip test.** The hanging wire test was performed to assess muscle strength and coordination in all experimental groups of mice (CUP, CUP+FES, CON, and CON+FES) subjected to cuprizone-induced demyelination and/or FES treatment. The apparatus consisted of a horizontal steel wire (80 cm in length, 7 mm in diameter) suspended between two platforms. Mice were habituated to the wire one day prior to testing. On the test day, each mouse was allowed to hang from the wire for three trials, with a 5-minute rest period between trials. The time until the mouse fell was recorded for each trial, and the average of the three measurements was calculated as the endurance time for each mouse [26].

**Shuttle box test.** The shuttle box test was used to evaluate learning and memory performance in all experimental groups (CUP, CUP+FES, CON, and CON+FES) of mice. The apparatus consisted of a two-compartment box, one illuminated and one dark, separated by a guillotine door. During the training session, each mouse was placed in the illuminated compartment, and upon entering the dark compartment, a mild foot shock (0.5 mA, 2 s) was delivered. Mice

were returned to their home cages after training. Memory retention was assessed 24 hours later by placing the mouse again in the illuminated compartment and recording the latency to enter the dark compartment, without delivering a shock. The longer latency to enter the dark compartment was considered indicative of better memory performance [27].

## Cytokines measurement by ELISA reader

Hippocampus and cerebellum tissues were collected from all experimental groups (CON, CON+FES, CUP, and CUP+FES; n = 3 per group for ELISA analysis, randomly selected) after completion of behavioral testing, which was conducted 30 days after the end of the cuprizone diet. The tissues were homogenized in a combination of phosphate-buffered saline (PBS) and RIPA buffer, then centrifuged at 30,000 rpm to obtain a clear supernatant. This supernatant was used for quantification of TNF and IL-1β levels. Commercial ELISA kits (Karmania Pars Gene Co., Kerman, Iran) were applied according to the manufacturer's instructions. To generate the standard curve, 500 μL of the sterile solution was added to each well and incubated at room temperature for 25 minutes. Subsequently, 20 μL of stop solution was added to each well, and the absorbance was read at 455 nm using a microplate reader. Cytokine concentrations were calculated based on the established standard curve [28].

## Real time PCR

Hippocampus and cerebellum tissues were collected from all experimental groups (CUP, CUP+FES, CON, and CON+FES; n = 4 per group for PCR analysis, randomly selected) to evaluate the expression of MBP and Olig2 genes (Table 1). Approximately 20–30 mg of tissue was homogenized in a mixture of 500 μL phosphate-buffered saline (PBS) and 500 μL RIPA buffer. Total RNA was extracted, and cDNA synthesis was performed according to the manufacturer's protocol (Karmania Pars Gene Co., Kerman, Iran). Relative gene expression was analyzed using a Real-Time PCR Master Mix containing 5 μL SYBR Green, 3 μL DEPC-treated water, 1 μL activator, and 1 μL cDNA template. Amplification was conducted on a Rotor-Gene device with cycling conditions of 95°C for 10 seconds and 60°C for 30 seconds. The stability of the housekeeping genes (GAPDH and HPRT) was measured across all experimental groups, and GAPDH, showing superior stability, was used as the reference gene for relative quantification. Gene expression levels were calculated using the $2^{-\Delta\Delta Ct}$ method relative to the housekeeping gene [29].

## Luxol Fast Blue (LFB) staining

Cerebellar tissues from all experimental groups were fixed in 4% paraformaldehyde (PFA), embedded in paraffin, and cut into 5 μm sections. Sections were deparaffinized using xylene and rehydrated through a descending ethanol series (100%, 95%, 70%) to reach distilled water. The slides were then incubated in Luxol Fast Blue solution at 56°C overnight. After staining, sections were rinsed in 95% ethanol and water, then differentiated in 0.05% lithium carbonate ($Li_2CO_3$) for 5 minutes, followed by 70% ethanol until the white and gray matter were clearly distinguishable. Finally, sections were

**Table 1. Primer sequences used for quantitative real-time PCR analysis.**

| Gene | Primer type | Sequence (5'→3') | Length (nt) | Tm (°C) |
|------|-------------|------------------|-------------|---------|
| GAPDH | Forward | TGACCTCAACTACATGGTCTACA | 23 | 60.2 |
| GAPDH | Reverse | CTTCCCATTCTCGGCCTTG | 19 | 60.2 |
| MBP | Forward | TCACAGCGATCCAAGTACCTG | 21 | 61.5 |
| MBP | Reverse | CCCCTGTCACCGCTAAAGAA | 20 | 61.5 |
| Olig2 | Forward | GGGAGGTCATGCCTTACGC | 19 | 62.5 |
| Olig2 | Reverse | CTCCAGCGAGTTGGTGAGC | 19 | 62.5 |

dehydrated, cleared with xylene, and mounted. Myelinated fibers appeared blue, whereas demyelinated regions remained pale when examined under a light microscope.

## Quantification of myelination

The degree of myelination was quantified on LFB-stained sections using ImageJ software (NIH, USA). For each animal, three cerebellar sections were selected for analysis. Within each section, four equally sized, non-overlapping regions of interest (ROIs) were systematically selected at predefined anatomical locations in the white matter, and quantification was performed blind to the experimental group to reduce sampling bias. The mean gray value of each ROI was measured from the red channel of the RGB image, which provides optimal contrast for LFB staining. Background correction was performed by analyzing a separate tissue-free image using the same approach. Relative myelin intensity for each sample was calculated as a percentage using the established formula.

$$100 \times \left( 1 - \frac{_{sample}\text{Mean Intensity}}{_{background}\text{Mean Intensity}} \right) = (\%) \text{ Relative Myelin Intensity}$$

The four ROIs per section were averaged to represent the section's value, and the mean±SEM was calculated from three sections per group. Higher percentage values correspond to stronger LFB staining and greater myelin content, while lower values indicate demyelination [30].

## Statistical analysis

Statistical analysis was performed using GraphPad Prism (version 10, GraphPad Software, USA). Data are presented as mean±SEM. Comparisons among multiple groups were conducted using one-way analysis of variance (ANOVA) followed by Tukey's post hoc test. A p-value of less than 0.05 was considered statistically significant.

## Results

### CUP reduced locomotor distance, velocity, and mobility without affecting inner zone preference, whereas FES facilitated partial functional recovery

In the open field test, total locomotor activity, assessed by the total distance traveled, was significantly lower in the CUP group compared with the CON group ($p < 0.001$) and the CON+FES group ($p < 0.01$). Additionally, significant differences were observed between the CON+FES group and both the CUP group ($p < 0.001$) and the CUP+FES group ($p < 0.01$) (Fig 1a). Velocity, used as an additional measure of spontaneous motor activity, was also significantly decreased in the CUP group compared with controls ($p < 0.001$). Administration of CUP+FES significantly increased velocity relative to the CUP group ($p < 0.001$). Moreover, velocity in the CON+FES group was significantly higher than in both CUP and CUP+FES groups ($p < 0.001$) (Fig 1b). Analysis of inner zone duration, reflecting time spent in different areas of the arena, revealed no significant differences among the groups (Fig 1c). Mobility duration, an indicator of overall movement engagement, was significantly reduced in the CUP group compared with controls ($p < 0.001$). The CON+FES group showed greater mobility duration relative to the CUP group ($p < 0.01$). Importantly, FES significantly improved cuprizone-induced impairments, as the CUP+FES group exhibited a significant increase in mobility duration compared with the CUP group ($p < 0.01$) (Fig 1d).

### CUP induces severe motor weakness and coordination deficits, while FES provides partial functional recovery

In the wire grip test, which assesses forelimb strength and neuromuscular endurance, the CUP group exhibited a markedly reduced grip strength compared with the CON group ($p < 0.001$). The CUP+FES group also showed significantly lower grip strength than controls ($p < 0.001$). Significant differences were observed between the CON+FES group and

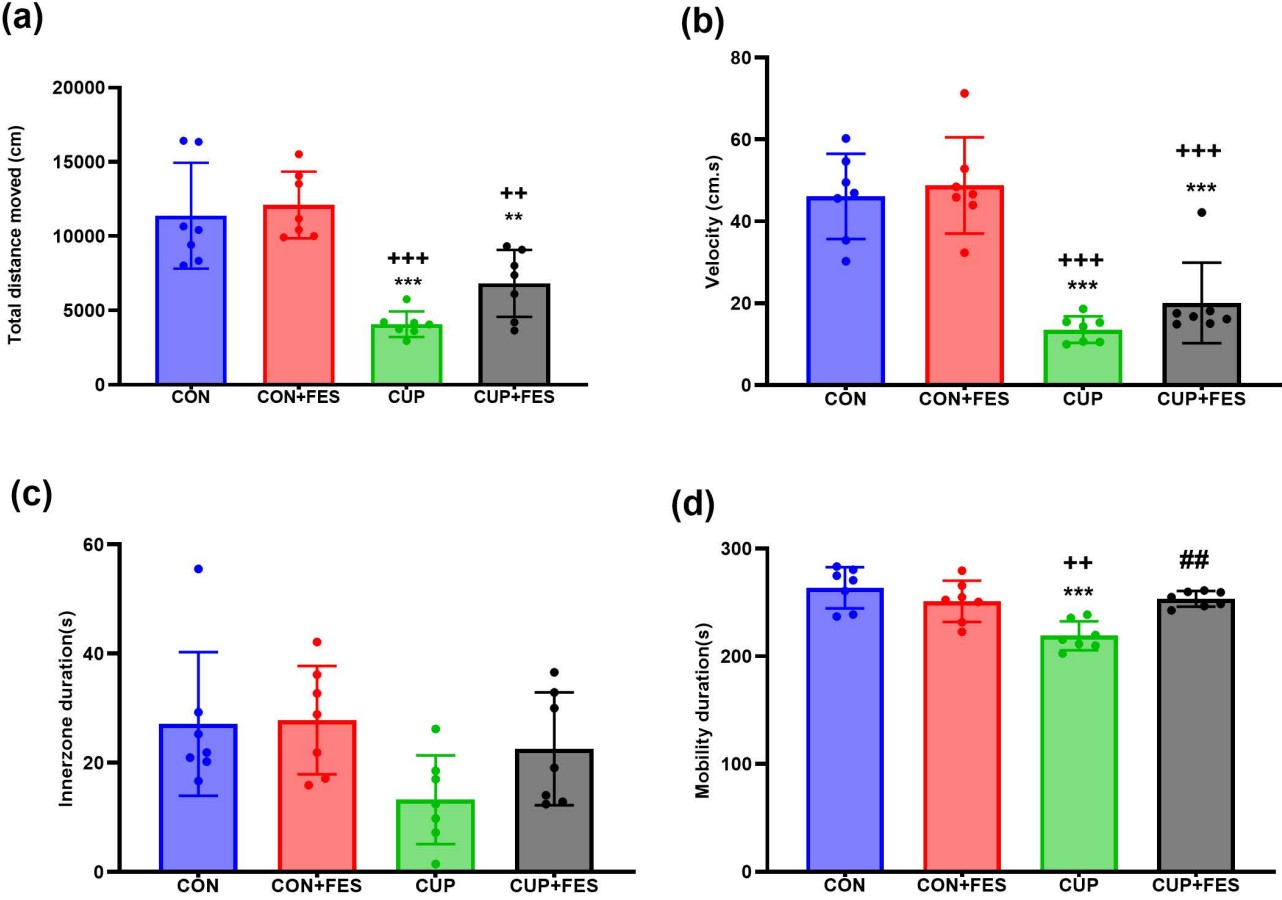

**Fig 1. CUP decreased activity levels, while FES improved performance in the open field test.** Symbols *, + and # indicate significant differences compared with the CON, CON+FES and CUP groups, respectively. Results are shown as mean±SEM.

both the CUP and CUP+FES groups (p<0.001). Despite the overall impairment, the CUP+FES group demonstrated a modest yet significant improvement compared with the CUP group (p<0.05), indicating improved functional performance (Fig 2a). Similarly, performance in the rotarod test, a measure of motor coordination and balance, was significantly reduced in the CUP group relative to controls (p<0.01). The CON+FES group showed significantly improved rotarod performance compared with the CUP group (p<0.001). Furthermore, a significant difference was detected between the CON+FES and CUP+FES groups (p<0.01), indicating superior motor coordination in the CON+FES condition (Fig 2b).

**FES partially restores passive avoidance learning deficits and reduces dark-compartment preference in CUP-treated mice**

A significant increase in the number of shocks was observed in the CUP group compared to the CON group (p<0.05), while no significant differences were noted among the other experimental groups (Fig 3a). In the shuttle box test, step-through latency (STL), an index of passive avoidance memory was significantly reduced in the CUP group compared with the CON group (p<0.001, Fig 3b). The CUP+FES group also demonstrated significantly lower STL values than the CON group (p<0.01). Additionally, the CON+FES group showed significantly improved STL performance relative to the CUP group (p<0.01) (Fig 3b).

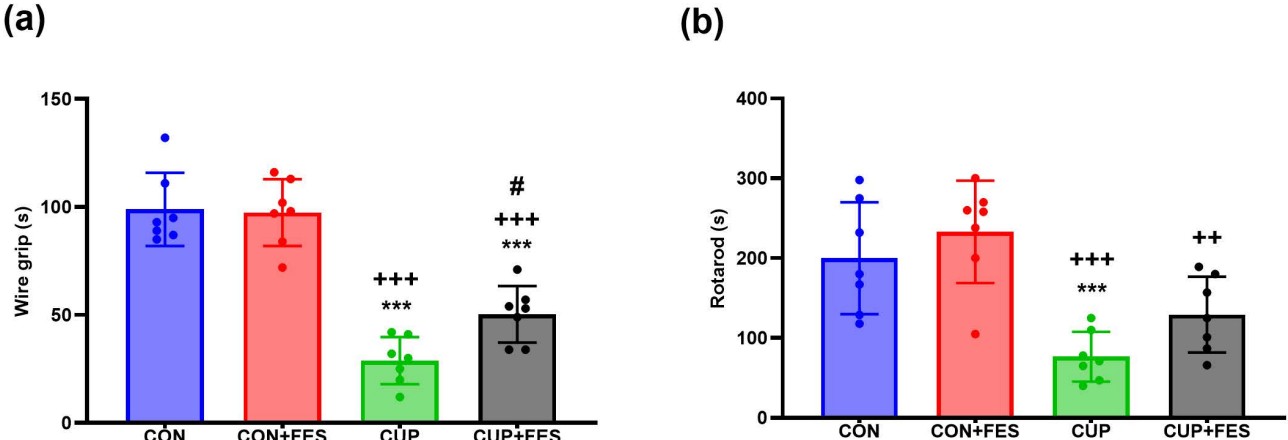

**Fig 2. Impact of various treatments on muscle strength and motor coordination assessed through wire grip and rotarod tests.** Symbols *, + and # indicate significant differences compared with the CON, CON+FES and CUP groups, respectively. Results are shown as mean±SEM.

Analysis of time spent in the dark compartment (TDS), reflecting memory retention and avoidance behavior, revealed a significant difference between the CON and CUP groups (p<0.001). The CON+FES group also exhibited significantly reduced TDS compared with the CUP group (p<0.001). Importantly, TDS values were significantly different between the CUP and CUP+FES groups (p<0.001), indicating improved behavioral performance following FES treatment (Fig 3c).

### ELISA analysis revealed that FES partially suppressed pro-inflammatory cytokine levels in CUP-treated mice

Cerebellar TNF expression was significantly elevated in the CUP group compared with the CON group (p<0.05). Similarly, the CON+FES group showed significantly lower TNF levels compared with the CUP group (p<0.05) (Fig 4a). In the hippocampus, TNF expression was markedly increased in the CUP group relative to controls (p<0.001). The CON+FES group also exhibited significantly lower TNF levels compared with the CUP group (p<0.001). Furthermore, TNF expression was significantly lower in the CUP+FES group than in the CUP group (p<0.01), indicating reduced neuroinflammation following FES treatment (Fig 4b).

Cerebellar IL-1β levels were significantly higher in the CUP group compared with the CON group (p<0.001). Significant reductions in IL-1β expression were observed in the CUP+FES group relative to controls (p<0.01), in the CON+FES group compared with the CUP group (p<0.001), and in the CON+FES group compared with the CUP+FES group (p<0.01) (Fig 4c). Similarly, hippocampal IL-1β expression was markedly elevated in the CUP group compared with the CON group (p<0.001). The CON+FES group demonstrated significantly lower IL-1β expression relative to the CUP group (p<0.001) (Fig 4d).

### FES treatment partially improves MBP and Olig2 expression levels in the cerebellum and hippocampus

MBP expression exhibited significant group-dependent alterations in both the cerebellum and hippocampus. In the cerebellum, MBP levels were markedly reduced in the CUP group compared to CON (p<0.001), with significant differences observed between CON and CUP+FES, CON+FES and CUP, and CON+FES and CUP+FES (all p<0.001). Notably, FES treatment significantly increased MBP expression in CUP+FES compared with CUP (p<0.05), indicating increased myelin-associated expression (Fig 5a). A similar pattern was observed in the hippocampus, where MBP levels were significantly decreased in CUP relative to CON (p<0.001). Treatment with FES markedly elevated MBP expression in CUP+FES compared with CUP (p<0.001), and comparisons involving CON, CON+FES, CUP, and CUP+FES showed

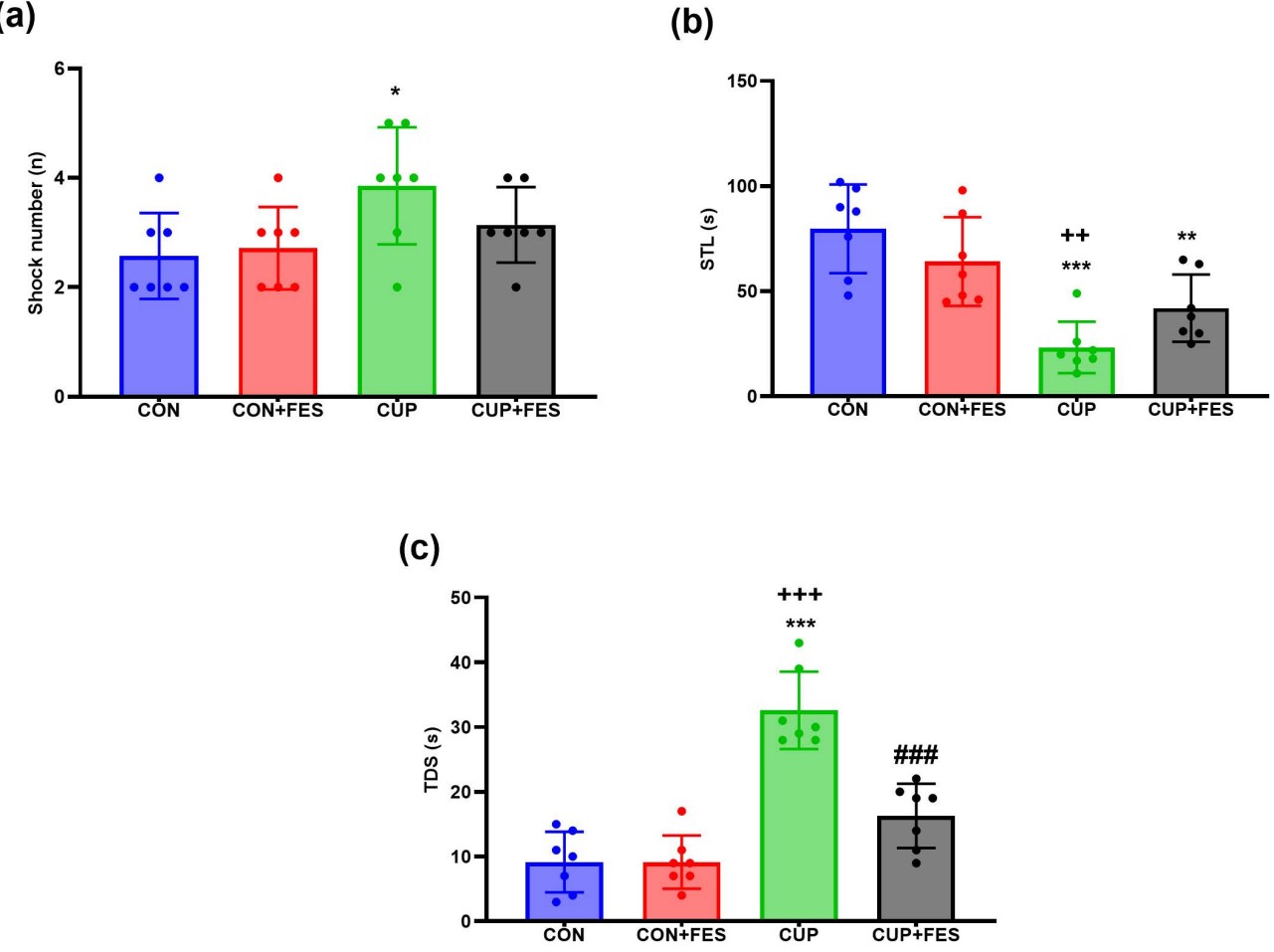

**Fig 3. Effect of different treatments on learning and memory in the shuttle box test.** Symbols *, + and # indicate significant differences compared with the CON, CON+FES and CUP groups, respectively. Results are shown as mean±SEM.

highly significant differences (p<0.001). Overall, these findings demonstrate that FES increases MBP expression following cuprizone-induced demyelination in both regions (Fig 5b).

Olig2 expression showed clear group-dependent alterations in both the cerebellum and hippocampus. In the cerebellum, a modest but significant difference was observed between CON and CON+FES (p<0.05), indicating slight modulation of baseline Olig2 levels by FES. Cuprizone exposure markedly reduced Olig2 expression in CUP compared with CON (p<0.001), and CUP+FES also differed significantly from CON (p<0.001). Olig2 levels in CON+FES were significantly different from both CUP and CUP+FES (p<0.001). Importantly, FES treatment significantly increased Olig2 expression in CUP+FES compared with CUP (p<0.01), indicating increased oligodendroglial lineage–associated expression (Fig 5c). A similar pattern was observed in the hippocampus, where CON and CON+FES showed a small but significant difference (p<0.05). CUP markedly altered Olig2 expression relative to CON (p<0.001), and CUP+FES also differed significantly from CON (p<0.001). Likewise, CON+FES showed strong differences compared with both CUP and CUP+FES (p<0.001). Notably, Olig2 expression was robustly increased in CUP+FES compared with CUP alone (p<0.001), indicating enhanced oligodendroglial lineage activity in the hippocampus (Fig 5d).

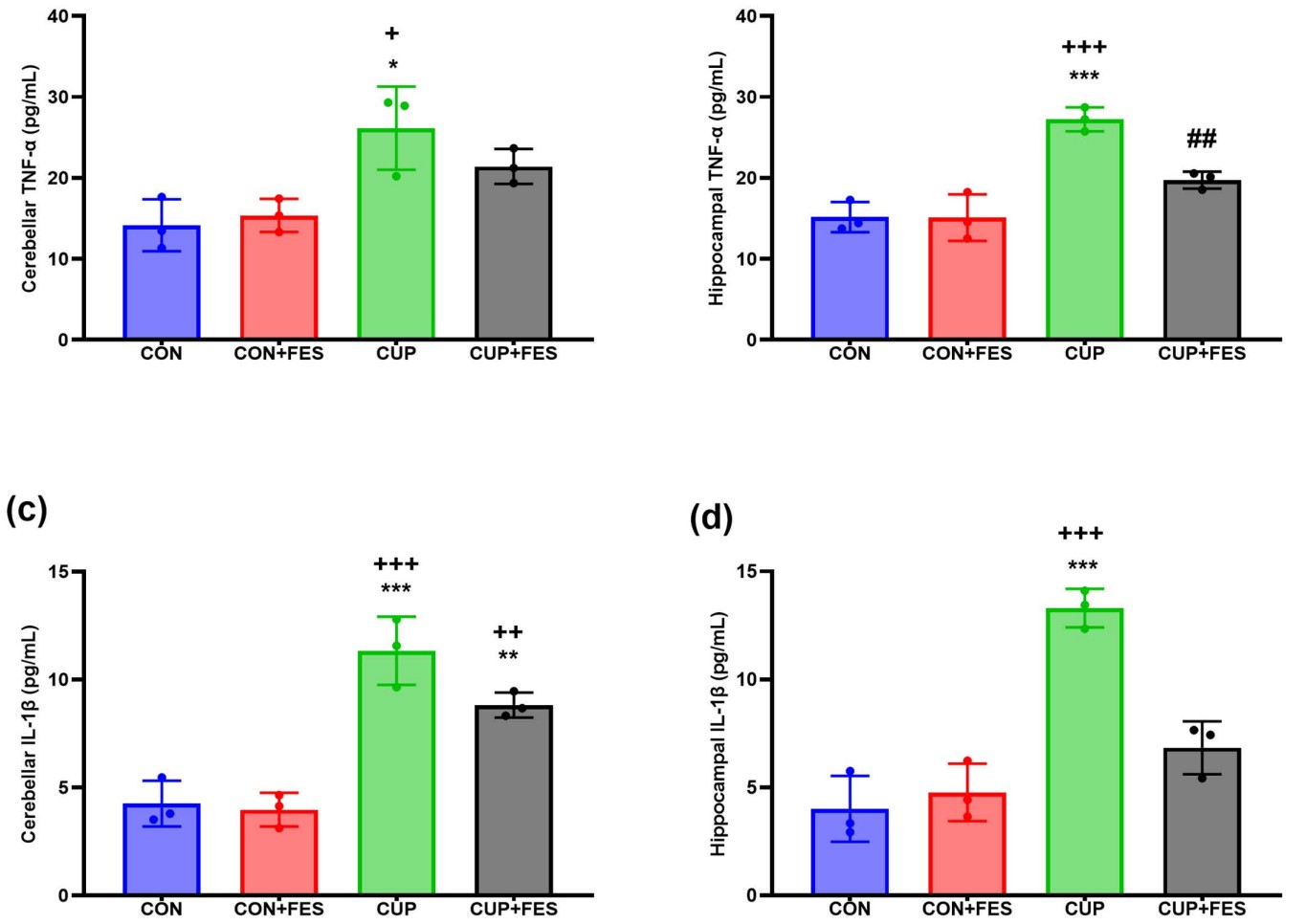

**Fig 4. TNF and IL-1β levels in cerebellum and hippocampus measured by ELISA.** Data are expressed as mean±SEM. Symbols *, + and # indicate significant differences compared with the CON, CON+FES and CUP groups, respectively.

## FES treatment effectively mitigates cuprizone-induced histopathological damage in brain tissue

Histopathological analysis showed significant differences between groups (p<0.001). Cuprizone administration induced marked demyelination compared to CON. Treatment with FES products (CUP+FES) resulted in significant attenuation of histopathological damage relative to CUP alone (p<0.001). Comparisons between CON+FES and both CUP and CUP+FES groups also revealed statistically significant differences (p<0.001), demonstrating the protective effects of FES. These findings indicate that FES treatment effectively mitigates cuprizone-induced histopathological alterations (Fig 6).

## Discussion

The findings of this study demonstrate that treatment with FES in cuprizone-exposed mice leads to clear improvements across multiple levels of analysis. Behavioral tests showed noticeable functional recovery in the CUP+FES group compared with the CUP group; however, statistical analyses indicated that, despite this improvement, the CUP+FES group

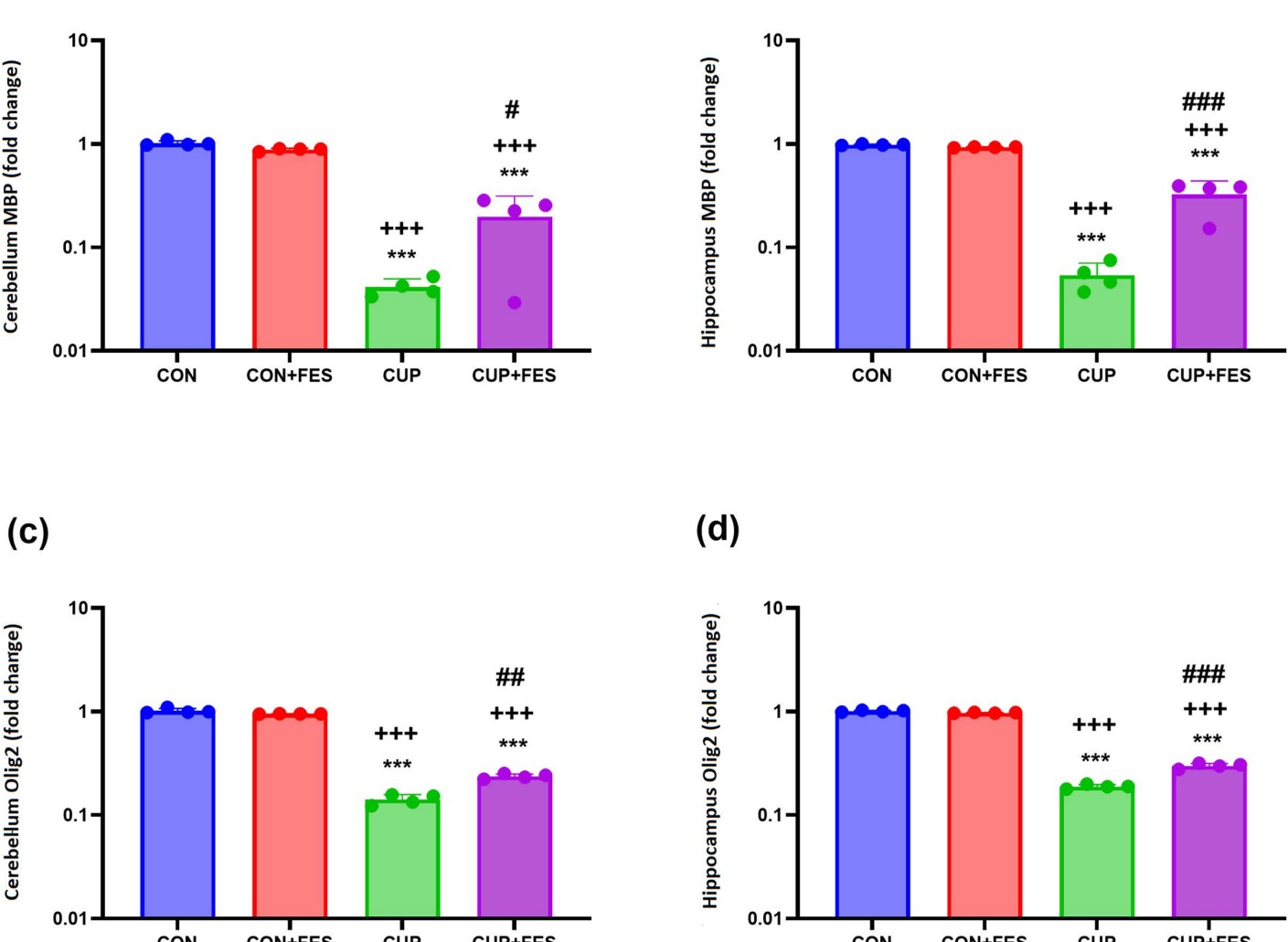

**Fig 5. Relative mRNA expression (fold change) of MBP and Olig2 in the hippocampus and cerebellum measured by PCR.** Data are expressed as mean±SEM. Symbols *, + and # indicate significant differences compared with the CON, CON+FES and CUP groups, respectively. Final analysis was performed by GAPDH.

remained considerably below the performance levels of both control groups. In addition, the CON and CON+FES groups exhibited largely comparable outcomes across most behavioral and molecular parameters, although some differences were observed, suggesting that FES alone may exert modest effects under baseline conditions. Molecular assessments, including both ELISA and PCR performed in the hippocampus and cerebellum, showed reduced TNF and IL-1β levels and increased expression of the myelination-related genes MBP and Olig2 in the CUP+FES group compared with the CUP group. Consistent with these findings, Luxol Fast Blue staining confirmed enhanced myelination in the treatment group. Overall, these results collectively indicate that FES is associated with improvements in myelin-related outcomes, while also highlighting that the extent of recovery in the CUP+FES group remains significantly below control levels.

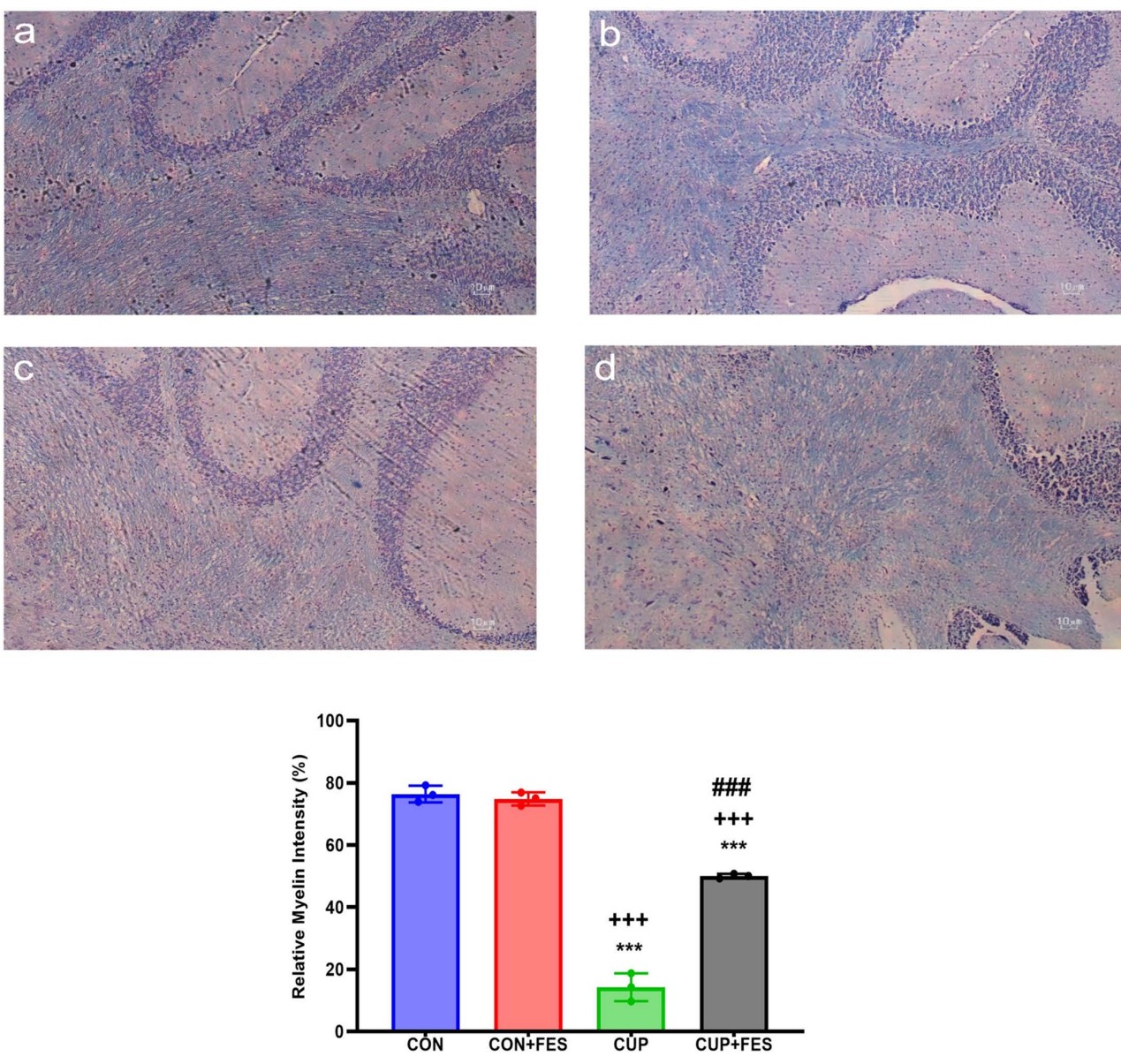

**Fig 6. Luxol Fast Blue staining was used to assess myelin levels.** Representative histopathological images. Groups are as follows: a, CON; b,CON+FES; c, CUP; d, CUP+FES. In the graph below, data are expressed as Mean±SEM. Symbols *, + and # indicate significant differences compared with the CON, CON+FES and CUP groups, respectively.

The observed enhancement of myelin-related outcomes in the cuprizone model following treatment with FSP aligns with the growing body of literature on the potent immunomodulatory capabilities of helminth-derived molecules. Our findings are consistent with the notion that these molecules can suppress deleterious neuroinflammation, a key driver of demyelination. For instance, a study by Maria E. Lund et al. [31] demonstrated that a parasite-derived peptide could ameliorate disease in murine models of MS, directly supporting the concept that helminth-derived factors have therapeutic potential in neuroinflammatory contexts.

The ability of FES to reduce neuroinflammation and support myelin preservation in our study is consistent with reports that certain *Fasciola hepatica* proteins, such as GST, can suppress inflammatory pathways including NF-κB [32]. This may contribute to the reduced oligodendrocyte apoptosis and improved myelination observed in our study. However, our findings should be interpreted within the context of prior immunological studies. For instance, Dowling et al. [33] showed that *Fasciola hepatica* secretory antigens induce a suppressive dendritic cell phenotype that attenuates Th17 responses, supporting their anti-inflammatory properties. additionally, our data demonstrate enhanced remyelination, suggesting that FES may exert both immunomodulatory and pro-regenerative effects beyond those identified in simplified immune systems. The study by Bąska et al. [34] reported minimal or undetectable effects of *Fasciola hepatica* excretory–secretory products on miRNA expression in THP-1 macrophages. In contrast, our in vivo findings show significant anti-inflammatory and pro-remyelinating effects, likely due to the greater complexity of multicellular interactions in vivo compared to simplified in vitro systems. The complex interactions present within the intact CNS in the cuprizone model may reveal therapeutic effects of FES that are not detectable in simplified monocytic or dendritic cell cultures [35]. The crucial difference between the complexity of in vivo systems and the reductionism of in vitro studies is a recurring theme in parasitology research, as the immunomodulatory effects of helminth-derived products often become fully apparent only within an integrated physiological context [36,37]. Reductionist systems, such as THP-1 macrophages or isolated dendritic cell assays, lack the integrated inflammatory and regenerative networks essential for remyelination, potentially explaining the minimal effects on Th2 activation or miRNA profiles observed in studies such as that by Dowling et al. and those by Bąska et al. Factors such as dosing differences, systemic administration versus direct culture exposure, assay sensitivity, and variability in ES composition (e.g., GST proteins, peptides, or whole ES mixtures) contribute to these discrepancies. The route and context of exposure are critical, as the same helminth-derived molecule can elicit different effects on immune cell populations in vivo, which cannot be replicated by single cell-type cultures [38]. When evaluated in complex in vivo disease models such as septic shock, EAE (experimental autoimmune encephalomyelitis), or cuprizone-induced demyelination, helminth-derived products consistently demonstrate anti-inflammatory and disease-modifying effects, as highlighted in the aforementioned studies. Conversely, limited or negative outcomes in isolated cell-based assays may stem from factors such as assay sensitivity, cell-type specificity, or the likelihood that the primary modulatory effects of FSPs occur at the cytokine and signaling levels, rather than through significant miRNA alterations detectable via microarray analysis.

The behavioral assessments in the cuprizone model demonstrated clear deficits in locomotion, coordination, and memory functions following demyelination induction. These impairments are consistent with disrupted neural conduction due to myelin loss, impacting motor and cognitive circuits. Treatment with FES led to significant improvements in motor performance, including enhanced locomotor activity and coordination, as shown in open field, rotarod, and wire grip tests. Additionally, FES partially restored memory function, as demonstrated by improved passive avoidance learning, although the recovery of learning itself was not as pronounced. Overall, behavioral tests showed partial improvement, indicating functional restoration with some remaining deficits. These functional recoveries are consistent with FES-associated changes in myelination.

ESP significantly reduced pro-inflammatory cytokines, including TNF and IL-1β, as evidenced by ELISA assays. This reduction is consistent with a shift toward a less inflammatory environment [39,40]. While these findings support an anti-inflammatory effect of ESPs, the downstream cellular and molecular consequences should be interpreted with caution. The observed increase in Olig2 and MBP expression may suggest an association with oligodendrocyte lineage activity and myelin-related processes, rather than definitively demonstrating enhanced OPC recruitment or myelin synthesis. Given the scope of the present data, mechanistic inferences regarding specific signaling pathways remain speculative [41]. Instead, our findings are more appropriately interpreted as evidence of an anti-inflammatory milieu that may be permissive for endogenous repair processes [42,43]. Similarly, increased MBP levels should be interpreted as indicative of changes in myelin-associated protein expression, without overattributing this to definitive remyelination or functional restoration. While these molecular changes are consistent with the direction of the observed behavioral improvements,

a direct causal relationship cannot be conclusively established based on the current dataset. Luxol Fast Blue staining of the cerebellar white matter confirmed enhanced myelin content in FES-treated mice compared with untreated cuprizone controls. This histological observation supports the presence of myelin-associated changes, although it should be interpreted alongside molecular findings with appropriate caution, collectively suggesting a potential association between FES treatment and myelin-related changes rather than definitively establishing remyelination and functional restoration.

Several limitations in the current study should be acknowledged when interpreting the findings. While the cuprizone model effectively induces demyelination, it lacks the autoimmune components characteristic of multiple sclerosis, limiting its ability to fully replicate conditions driven by adaptive immunity and to predict the effects of FES in such scenarios. The cuprizone model was therefore chosen over the more commonly used EAE model because it produces highly reproducible toxin-mediated demyelination that is independent of peripheral adaptive immune responses. Additionally, the use of a crude *Fasciola hepatica* excretory–secretory (FES) preparation precludes identification of the specific molecule(s) responsible for the observed effects. While FES treatment resulted in significant improvements in motor and cognitive function, cytokine levels, and myelin markers, the extent of recovery remained partial compared with control animals. Furthermore, the exclusive use of male mice limits the generalizability of our findings, given established sex differences in neuroinflammatory responses and remyelination capacity. Finally, the lack of more advanced mechanistic studies, such as single-cell transcriptomics, electrophysiological recordings, or lineage-tracing analyses, restricts deeper insight into the cellular and molecular pathways involved. Future studies addressing these limitations will be important to further evaluate the therapeutic potential of helminth-derived products in multiple sclerosis.

## Conclusion

This study shows that *Fasciola hepatica* excretory–secretory products (FES) exert neuroprotective effects in the cuprizone model. FES treatment partially improved locomotor and cognitive performance, reduced pro-inflammatory cytokines in the hippocampus and cerebellum, and increased MBP and Olig2 expression. Histological analysis revealed better myelin preservation in treated animals. These findings indicate that FES attenuates neuroinflammation and supports myelin preservation and oligodendrocyte lineage activation rather than inducing robust remyelination. While recovery was incomplete, the results suggest that helminth-derived molecules may help create a less inflammatory environment that favors myelin protection. Further studies in autoimmune MS models and with optimized treatment timing are warranted.

## Acknowledgments

We express our sincere appreciation to the Neuroscience Research Center of Kerman University of Medical Sciences for providing the facilities required for conducting this study. We further extend our gratitude to Dr. Fatemeh Shahsavari for her valuable participation in this project. We also acknowledge the use of ChatGPT (version 5) strictly for assistance with language editing and grammar.

## Author contributions

**Conceptualization:** Aliakbar Mariki, Mohammad Shabani.

**Investigation:** Aliakbar Mariki.

**Methodology:** Alireza Keyhani, Majid Fasihi Harandi, Mansoureh Sabzalizadeh.

**Supervision:** Mohammad Shabani.

**Validation:** Kristi Anne Kohlmeier, Seyed Mohammad Mousavi, Mohammad Shabani.

**Writing – original draft:** Aliakbar Mariki.

**Writing – review & editing:** Kristi Anne Kohlmeier, Seyed Mohammad Mousavi, Alireza Keyhani, Majid Fasihi Harandi, Mohammad Shabani.

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
