## [Decision Letter · Decision Letter 0]

27 Feb 2026

PONE-D-26-04172Fasciola hepatica Excretory-Secretory Products Enhance Remyelination and Reduce Neuroinflammation in the Cuprizone –Induced Multiple Sclerosis ModelPLOS One

Dear Dr. Shabani,

Thank you for submitting your manuscript to PLOS ONE. After careful consideration, we feel that it has merit but does not fully meet PLOS ONE’s publication criteria as it currently stands. Therefore, we invite you to submit a revised version of the manuscript that addresses the points raised during the review process.

We look forward to receiving your revised manuscript.

Kind regards,

Tanja Grubić Kezele, Ph.D., M.D.

Academic Editor

PLOS One

**Journal Requirements:**

2.  We note that your Data Availability Statement is currently as follows:

“All relevant data are within the manuscript and its Supporting Information files.”

4. We note you have included a table to which you do not refer in the text of your manuscript. Please ensure that you refer to Table 1 in your text; if accepted, production will need this reference to link the reader to the Table.

**Additional Editor Comments:**

Based on the reviewers' suggestions, the paper needs major revision. The reviewers' comments can be found below.

Reviewers' comments:

Reviewer's Responses to Questions

**Comments to the Author**

1. Is the manuscript technically sound, and do the data support the conclusions?

Reviewer #1: Partly

Reviewer #2: Partly

2. Has the statistical analysis been performed appropriately and rigorously? 

Reviewer #1: Yes

Reviewer #2: I Don't Know

3. Have the authors made all data underlying the findings in their manuscript fully available?

Reviewer #1: Yes

Reviewer #2: Yes

4. Is the manuscript presented in an intelligible fashion and written in standard English?

Reviewer #1: No

Reviewer #2: Yes

5. Review Comments to the Author

Reviewer #1: Manuscript No. PONE-D-26-04172

Fasciola hepatica Excretory-Secretory Products Enhance Remyelination and Reduce Neuroinflammation in the Cuprizone-Induced Multiple Sclerosis Model.

General opinion.

This is an interesting work that demonstrate the therapeutic potential of F. hepatica ESP in ameliorating the MS-pathology in a mouse model. The manuscript has certain merit; however, it requires major modifications to be suitable for publication in PLOS One.

Major Comments

1. The number of animals per group was 7. Please provide evidence of power analysis.

2. Why two different concentrations of Cuprizone (0.7% w/w first and then 0.2% w/w) for inducing demyelination? The more frequent cuprizone concentration in standard rodent chow is 0.2%, which is associated with extensive and reproducible demyelination. Higher concentrations (0.4%-0.6%) are only necessary in aged mice, which is not the case because the mice were 6-week-old. Please provide a rationale for this.

3. Despite the excretory-secretory products are prepared in sterile conditions this does not guarantee that the antigen is endotoxin-free. How did you assured that antigens were endotoxin-free?

4. Please, provide a proper rationale for F. hepatica ESP amount used in injecting animals (10ug). In the reference cited (Lume ME et al 2014), the F. hepatica ESP were produced by culturing parasites for only 8 hours vs. 24 hours used for you, which definitively can lead to a change in the antigen mixture composition and thereby, the dose used by Lume in a diabetes model with NOD mice could not necessarily works in your cuprizone MS model. Why didn’t you assay a dose-response experiment? Why don’t you administer the antigen based on body weight instead a single a unique amount?

5. It would be recommendable to perform additional experiments aimed to determine how T-cells (Th1, Th17 and Treg) were affected by the treatment with F. hepatica ESP as well as also investigate the effect of antigen on cytokines associated to MBP. This is an autoantigen closely related to the activation of CD4+ T-cells, which produce high levels of IFNγ, IL-17 and IL-6. Alternatively, a western blot analysis of spinal cords or brain homogenates showing reduction of specific bands intensity (18.8, 17, or 14kDa) could support the PCR amplification analysis.

6. Why using a cuprizone-induced demyelination instead the experimental autoimmune encephalomyelitis (EAE) model, which is the commonly used? A rationale for this should be added and in the discussion pros and contrast of the cuprizone-intoxication model.

7. Some statements in the discussion seems to be inaccurate or overinterpreted. For example: lines 382-386, you state that in vivo effect of Fh ESP on remyelination are in contrast with the Dowling et al studies [28]. The meaning of this apparent discrepancy is not clear to me.

8. Lines 386-391: It is not clear to me relation that you make between the results of Baska et al [29] obtained with THP1 macrophages, and the miRNA expression with the data presented herein.

Minor comments

1. All scientific names should be in italic: i.e. Fasciola hepatica, please correct it throughout the text and references list.

2. Please, provide the Ethical approval number of protocol in the Material and Methods section

3. Please, revise introduction and correct or add adequate citations when necessary. For example, lines 70-73, mention diverse mechanisms used by ESPs (downregulation of pro-inflammatory pathways, inhibition of inflammasome activation, suppression of T and B cell responses, T-cell expansion and activation of M2-macrophages. However, only is cited the work of Valdes-Fernandez et al 2024, which was exclusively about GST and M2-type induced macrophages.

Lines 73-75: The publication of Cengiz ZT et al 2023, which is not available in PubMed, is about antioxidant enzymes and oxidative stress in human fascioliasis and it is not related to multiple sclerosis or autoimmune diseases

Lines 75-78: The publications of Musah-Eroje M 2018 and Ali SBG, 2021 are about F. hepatica immunomodulation and do not mention neuroprotective microglia phenotype or myelin repair.

Lines 82-83: You state that cuprizone is a no toxic model, which is incorrect. It is considered a toxic model. Cuprizone causes toxicity by triggering mitochondrial stress, resulting in severe oxidative stress.

Lines 83-84: Administered ESPs were evaluated….., seems has no sense. Please check syntaxis.

4. Please, also revise the references cited in the discussion, which seems to be wrongly cited. E.g., in line 375: The study by Maria E. et al [26] demonstrated that…… The reference #26 in the reference list corresponds to: Lund ME et al 2016.

5. Please add the Doi and PMCID/PMID number to all references listed.

6. Please revise grammatical and syntaxis throughout the entire text

7. Please correct the graphical abstract. After centrifugation, the arrow wrongly mark the pellet that contains all debris of FES instead the supernatant. The mouse seems to be drinking instead eating solid food (rodent chow).

Reviewer #2: This manuscript describes an investigation into the capacity for the excretory/secretory products of Fasciola hepatica to prevent progression of neurodegenerative disease in a murine model of cuprizone induced demyelination. This study builds on the body of previous literature showing the potent beneficial effect of helminth infection and their secreted products to slow/stop the progression of immune mediated disease. The study follows a standard approach of treatment and assessment of outcomes using acceptable technical approaches.

I would suggest the conclusions and title of the paper are a little exaggerated based on the actual experimental data presented. While there is clearly an effect of the injection of FhES, there is no robust measure of remyelination per se. Accordingly, I think the conclusion should be tempered and the title edited to better reflect the experimental design and actual outcomes. In the Cuprizone model, while demyelination is certainly underway by week 2 (when the FhES is administered) this continues over the subsequent days if cuprizone is continually administered in the food up to 5 weeks before remyelination begins. In this study, the mice were on Cuprizone for during the same period as they received FhES. It is therefore more likely, that the FhES slowed down demyelination rather than induced remyelination – the increased expression of Olig2 being the only possible biomarker of oligodendrocyte activation, is not sufficient to conclude the induction of remyelination.

Additional comments:

• Line 65 – at the very least the seminal and foundational paper in this area by Correlae et al should be included here.

• Fasciola hepatica (and all other parasite names) should always be in italics.

• Line 66 – the work of Mills et al which explored the effect of ES in murine models of EAE should be included here

Methods:

• The reason for the sole choice of male mice should be included here

• What source was utilized for the protocol to harvest the Fasciola ES (which should be correctly named FhES)? 24h is a long time to culture these worms, with optimal secretion and work health being around 8h – after that the parasites being to break down. Giiven this, a protein gel of the harvested ES could be included in the paper as supplementary data.

• Why was this particular regime of FhES administration chosen? A reference is given, but this is from a very different murine model which runs over a much longer time frame. In addition, given the neurotropic disease under examination here, why was IV not chosen as an administration route?

• Why was the 10ug dose of ES chosen, and why was it diluted in PBS rather than saline?

• Was the stability of the housekeeping gene expression measured. This is critical if it is to be used as a relative marker to quantify the expression of other genes. This analysis needs to be completed and included in the paper.

• The quantification of myelination mentions choosing regions of interest – this suggests preferred regions of tissue were assessed. Ideally the same regions of cerebellar sections in all mice should be chosen and quantified blind.

Results

• TNF is no longer called ‘alpha’ due to the recharacterization of TNFbeta as LT – please correct this terminology throughout

• There is no explanation here or in the methods of the statistical analyses that were used to calculate the significance of any data – as such this makes it very difficult to properly interpret the outcomes.

• I would not use the term ‘partial suppression’ – either there was a reduction or not of cytokines in response to the FhES – the extent of this suppression can then be determined by statistical analyses.

• Why is the PCR gene expression data presented as a ratio in Figure 5 when in the methods it describes the quantification as calculated using house keeping gene?

• Can the authors clarify how many animals were used to calculate the data shown in Figs 4 and 5. Although the methods describe the use of 7 mice per group there appears to only be 3-4 represented in these data sets?

• Line 310 – to claim the ES restores MBP and Olig2 expression is a very exaggerated and wholly inaccurate when looking at the actual data that is presented. Levels of neither gene returns to the expression seen in the control mice which would represent a restoration. The language and conclusion from this data set needs to be tempered.

Discussion

• The discussion is far too long and takes too may exaggerated explorations into likely mechanisms given the small set of basic data that is presented here. It is best to edit this to a consideration of this study of ES efficacy compares to others (models, dose, read outs etc) – then make a unifies proposal for mechanism based on the evidence from this and the existing body of literature. (something like that presented in lines 382-412.

• The content of lines 425-454 is too hypothetical and makes assumptions incorrectly based on an exaggerated interpretation of the MBP and Olig2 expression data. This type of claim is not acceptable.

• Line 375 – the author of this citations [26] is Lund et al

• Line 377 – there is no basis for this mechanistic connection between ref #27 and the data that is shown here

• Lines 455 – this section is predominantly a reiteration of what was mentioned before – although recognizing some of the limitations is a positive move.

• The conclusion incorrectly states the induction of remyelination (please see my previous comments) there is just not enough actual evidence to make this claim.

6. PLOS authors have the option to publish the peer review history of their article (what does this mean?). If published, this will include your full peer review and any attached files.

Reviewer #1: No

Reviewer #2: No

---

## [Author Response · Author response to Decision Letter 1]

16 Apr 2026

Kerman University of Medical Sciences

Date: 04/12/2026

Ref: Submission ID: PONE-D-26-04172 - [EMID:338d89f5095101b8]

Resubmission of Manuscript:” Fasciola hepatica Excretory-Secretory Products Enhance Remyelination and Reduce Neuroinflammation in the Cuprizone –Induced Multiple Sclerosis Model.”

Dear Editor,

We are pleased to submit our manuscript with revisions addressing the reviewers’ comments. All responses have been provided in blue text.

We trust that the revisions adequately address the concerns raised and meet the required standards.

Yours sincerely,

M. Shabani

Mohammad Shabani, PhD

Neuroscience Research Center, Neuropharmacology institute, Kerman University of edical Sciences, Kerman, Iran.

P.O. Box: 76198-13159; Fax: +98-343-226-4198

E-mail: shabanimoh@yahoo.com

Editorial comments

Comments 1: Please ensure that your manuscript meets PLOS ONE's style requirements, including those for file naming. The PLOS ONE style templates can be found at

• Response to comment: We have carefully revised the manuscript to ensure full compliance with PLOS ONE style requirements. The file naming, manuscript structure, title page, and formatting have all been updated according to the journal’s guidelines and provided templates.

Comments 2: We note that your Data Availability Statement is currently as follows:

“All relevant data are within the manuscript and its Supporting Information files.”

• Response to comment: Thank you for your comment regarding data availability. We confirm that all raw data required to replicate the results of this study have been fully uploaded as Supporting Information files.

The files are provided and labeled as follows:

• Behavioral and Histopathology Supp (excel file): raw data for all behavioral experiments and final data of histopathology

• Molecular test Supp (excel file): raw molecular data, including ELISA and PCR (GAPDH AND HPRT)

• Histopathology Supp (word file): detailed histopathological results prior to analysis

Comments 3: Your ethics statement should only appear in the Methods section of your manuscript. If your ethics statement is written in any section besides the Methods, please move it to the Methods section and delete it from any other section. Please ensure that your ethics statement is included in your manuscript, as the ethics statement entered into the online submission form will not be published alongside your manuscript.

• Response to comment: The ethics statement has been removed from the Declarations section and has been incorporated into the Animal subsection of the Materials and Methods section.

Comments 4: We note you have included a table to which you do not refer in the text of your manuscript. Please ensure that you refer to Table 1 in your text; if accepted, production will need this reference to link the reader to the Table.

• Response to comment: The reference to Table 1 has been added to the Real-Time PCR section of the manuscript (first mention of the target genes.)

Reviewers' comments

Reviewer 1

We sincerely thank the first reviewer for her/his positive opinion on our research.

Major Comments

Comments 1: The number of animals per group was 7. Please provide evidence of power analysis.

• Response to comment: Each experimental group included 7 animals. A post hoc power analysis, based on hippocampal MBP expression (a primary outcome measure), revealed a large effect size (Cohen’s d ≈ 4.6) and confirmed adequate statistical power (power ≈ 0.99, α = 0.05). This example is representative of other outcome measures, which showed similarly high power values. These details, provided in response to your comment, are not typically reported but can be incorporated into the Methods section upon request.

Comments 2: Why two different concentrations of Cuprizone (0.7% w/w first and then 0.2% w/w) for inducing demyelination? The more frequent cuprizone concentration in standard rodent chow is 0.2%, which is associated with extensive and reproducible demyelination. Higher concentrations (0.4%-0.6%) are only necessary in aged mice, which is not the case because the mice were 6-week-old. Please provide a rationale for this.

• Response to comment: Our cuprizone dosing schedule, 0.7% (w/w) for the first week, followed by 0.2% (w/w) for weeks 2–4 was adopted from the validated protocol of Abd El Aziz et al. (2021) [16]. This two-phase approach, shown to induce robust demyelination and associated motor and cognitive deficits in young adult C57BL/6 mice (matching our age and strain), was selected for the following reasons: The initial 0.7% dose accelerates oligodendrocyte apoptosis, ensuring rapid demyelination onset in the hippocampus and cerebellum. The subsequent reduction to 0.2% sustains demyelination while mitigating high-dose toxicity, weight loss, and mortality, improving animal welfare and reproducibility. While 0.2% is typical for corpus callosum demyelination, our focus on the hippocampus and cerebellum, which are less susceptible to cuprizone, necessitated the two-phase regimen within our 4-week timeframe. This protocol reliably yielded significant pro-inflammatory cytokine elevation (TNF-α and IL-1β), reduced MBP and Olig2 expression, myelin degradation on Luxol Fast Blue staining, and locomotor, coordination, and memory deficits, validating its suitability for our study.

• We have clarified this rationale and the reference to the original validation study in the revised Materials and Methods section (Induction of Demyelination subsection).

Comments 3: Despite the excretory-secretory products are prepared in sterile conditions this does not guarantee that the antigen is endotoxin-free. How did you assured that antigens were endotoxin-free?

• Response to comment: Although the Fasciola hepatica excretory-secretory products (FES) were prepared under sterile conditions, we acknowledge that sterility alone does not guarantee the absence of endotoxin (LPS) contamination. To minimize potential endotoxin levels, the collected supernatant was centrifuged, and the final FES preparation was filtered through a 0.22 µm sterile filter before protein quantification and storage. The dose used (10 µg per injection) is low and consistent with previous in vivo studies using F. hepatica ESPs.

• Importantly, the observed effects of FES in the cuprizone model (reduced pro-inflammatory cytokines, increased MBP and Olig2 expression, and partial behavioral recovery) are opposite to those expected from endotoxin contamination, which typically exacerbates neuroinflammation and demyelination.

• We have now added the following sentence to the “Preparation of Fasciola hepatica Excretory-Secretory Products” subsection in the Materials and Methods: (To further reduce the risk of bacterial contamination, the final ES supernatant was filtered through a 0.22 µm sterile syringe filter prior to use.)

Comments 4: Please, provide a proper rationale for F. hepatica ESP amount used in injecting animals (10ug). In the reference cited (Lume ME et al 2014), the F. hepatica ESP were produced by culturing parasites for only 8 hours vs. 24 hours used for you, which definitively can lead to a change in the antigen mixture composition and thereby, the dose used by Lume in a diabetes model with NOD mice could not necessarily works in your cuprizone MS model. Why didn’t you assay a dose-response experiment? Why don’t you administer the antigen based on body weight instead a single a unique amount?

• Response to comment: The dose of 10 µg FES per injection was selected based on previous in vivo studies using F. hepatica excretory-secretory products in mouse models of autoimmune and inflammatory diseases, including the cited reference [18]. Although our ESP preparation involved a 24-hour collection period (compared to 8 hours in Lund et al.), the total protein concentration was carefully quantified by Bradford assay, and the final dose remained within the range reported as safe and effective in similar helminth-derived product studies.

• A dose-response experiment was not performed because this was an initial proof-of-concept study aimed at evaluating whether F. hepatica ESP possesses any remyelinating and anti-inflammatory potential in the cuprizone model. The fixed 10 µg dose (in 100 µL PBS) was chosen for practical reasons and to maintain consistency across animals, as all mice were of similar age and body weight (20–25 g). Administering a fixed amount rather than adjusting per body weight is common in such short-term mouse studies when weight variation is minimal.

Comments 5: It would be recommendable to perform additional experiments aimed to determine how T-cells (Th1, Th17 and Treg) were affected by the treatment with F. hepatica ESP as well as also investigate the effect of antigen on cytokines associated to MBP. This is an autoantigen closely related to the activation of CD4+ T-cells, which produce high levels of IFNγ, IL-17 and IL-6. Alternatively, a western blot analysis of spinal cords or brain homogenates showing reduction of specific bands intensity (18.8, 17, or 14kDa) could support the PCR amplification analysis.

• Response to comment: We thank the reviewer for this valuable suggestion. Due to limited financial resources and the scope of this initial proof-of-concept study, we were unable to perform additional experiments such as flow cytometry analysis of T-cell subsets (Th1, Th17, Treg) or measurement of cytokines specifically associated with MBP-reactive CD4+ T-cells (e.g., IFNγ, IL-17, IL-6). Similarly, Western blot analysis of MBP isoforms in spinal cord or brain homogenates could not be conducted. We agree that these experiments would provide deeper mechanistic insight into the immunomodulatory effects of F. hepatica ESP. Such analyses will be considered in future studies with adequate funding.

Comments 6: Why using a cuprizone-induced demyelination instead the experimental autoimmune encephalomyelitis (EAE) model, which is the commonly used? A rationale for this should be added and in the discussion pros and contrast of the cuprizone-intoxication model.

• Response to comment: The cuprizone-induced demyelination model was deliberately chosen over the experimental autoimmune encephalomyelitis (EAE) model for the following reasons:

• It allows the study of direct effects on remyelination and oligodendrocyte biology without the confounding influence of peripheral adaptive immune responses.

• It produces highly reproducible demyelination in specific brain regions (hippocampus and cerebellum), which are the focus of our behavioral and histological analyses.

• It is technically simpler, less variable, and requires fewer animals compared to EAE.

• Due to limited financial resources, we selected the cuprizone model as it is more cost-effective and better suited for our initial proof-of-concept investigation into the remyelinating potential of F. hepatica ESP.

We have added the following explanation to the Discussion section (final paragraph)

• While the cuprizone model effectively induces demyelination, it lacks the autoimmune components characteristic of multiple sclerosis, limiting its ability to fully replicate conditions driven by adaptive immunity and to predict the effects of FES in such scenarios. The cuprizone model was therefore chosen over the more commonly used experimental autoimmune encephalomyelitis (EAE) model because it produces highly reproducible toxin-mediated demyelination that is independent of peripheral adaptive immune responses. This allowed us to specifically evaluate the direct effects of F. hepatica ESP on oligodendrocyte precursor cell recruitment, remyelination, and local neuroinflammation without confounding autoimmune-driven T-cell responses.

Comments 7: Some statements in the discussion seems to be inaccurate or overinterpreted. For example: lines 382-386, you state that in vivo effect of Fh ESP on remyelination are in contrast with the Dowling et al studies [28]. The meaning of this apparent discrepancy is not clear to me.

• Response to comment: We have revised the text to clarify the interpretation in the third paragraph of the Discussion, beginning with “However, our findings should be interpreted…”.

Comments 8: Lines 386-391: It is not clear to me relation that you make between the results of Baska et al [29] obtained with THP1 macrophages, and the miRNA expression with the data presented herein.

• Response to comment: Following the previous comment, we have revised this section in the third paragraph of the Discussion (lines 407–412) to clarify the relationship between Bąska et al. [29] and our findings, emphasizing differences in experimental context (in vitro vs. in vivo).

• In the revised version it is Bąska et al. [33]

Minor comments

Comments 1: All scientific names should be in italic: i.e. Fasciola hepatica, please correct it throughout the text and references list.

• Response to comment: All scientific names, have been italicized throughout the manuscript and reference list.

Comments 2: Please, provide the Ethical approval number of protocol in the Material and Methods section

• Response to comment: The Ethical approval number of protocol has been provided in the Material and Methods section (Animal subsection)

Comments 3: Please, revise introduction and correct or add adequate citations when necessary.

• Response to comment: We have thoroughly revised the Introduction section to improve accuracy, clarity, and appropriate citation of the literature.

Comments 4: Please, also revise the references cited in the discussion, which seems to be wrongly cited. E.g., in line 375: The study by Maria E. et al [26] demonstrated that…… The reference #26 in the reference list corresponds to: Lund ME et al 2016.

• Response to comment: Unfortunately, the name of the first author was written incorrectly. In the revised version, we corrected it to Maria E. Lund et al.

Comments 5: Please add the Doi and PMCID/PMID number to all references listed.

• Response to comment: As indicated in the first editorial comment, all manuscript components, including references, were required to follow the journal’s guidelines. Accordingly, we downloaded the official PLOS ONE reference style from the journal’s website and applied it using EndNote.

Comments 6: Please revise grammatical and syntaxis throughout the entire text

• Response to comment: the manuscript has been carefully revised for grammar and syntax throughout the entire text

Comments 7: Please correct the graphical abstract. After centrifugation, the arrow wrongly mark the pellet that contains all debris of FES instead the supernatant. The mouse seems to be drinking instead eating solid food (rodent chow).

• Response to comment: The graphical abstract has been corrected according to the requested changes

Reviewer 2

We thank the second reviewer for his/her valuable feedback and constructive comments, which have helped us, improve the quality of our manuscript.

Comments 1: I would suggest the conclusions and title of the paper are a little exaggerated based on the actual experimental data presented. While there is clearly an effect of the injection of FhES, there is no robust measure of remyelination per se. Accordingly, I think the conclusion should be tempered and the title edited to better

---

## [Decision Letter · Decision Letter 1]

30 Apr 2026

PONE-D-26-04172R1Fasciola hepatica Excretory-Secretory Products Attenuate Demyelination and Reduce Neuroinflammation in the Cuprizone –Induced Multiple Sclerosis ModelPLOS One

Dear Dr. Shabani,

Thank you for submitting your manuscript to PLOS ONE. After careful consideration, we feel that it has merit but does not fully meet PLOS ONE’s publication criteria as it currently stands. Therefore, we invite you to submit a revised version of the manuscript that addresses the points raised during the review process.

We look forward to receiving your revised manuscript.

Kind regards,

Tanja Grubić Kezele, Ph.D., M.D.

Academic Editor

PLOS One

Journal Requirements:

**Additional Editor Comments:**

Your manuscript, entitled "Fasciola hepatica Excretory-Secretory Products Attenuate Demyelination and Reduce Neuroinflammation in the Cuprizone – Induced Multiple Sclerosis Model", has been reviewed. Your efforts to revise the manuscript are appreciated. However, the peer review process continues because Reviewer 2 requests further revisions on certain issues the author should address. Please find the reviewer's commentary below.

Reviewers' comments:

Reviewer's Responses to Questions

**Comments to the Author**

1. If the authors have adequately addressed your comments raised in a previous round of review and you feel that this manuscript is now acceptable for publication, you may indicate that here to bypass the “Comments to the Author” section, enter your conflict of interest statement in the “Confidential to Editor” section, and submit your "Accept" recommendation.

Reviewer #2: (No Response)

2. Is the manuscript technically sound, and do the data support the conclusions?

Reviewer #2: Yes

3. Has the statistical analysis been performed appropriately and rigorously? 

Reviewer #2: Yes

4. Have the authors made all data underlying the findings in their manuscript fully available?

Reviewer #2: Yes

5. Is the manuscript presented in an intelligible fashion and written in standard English?

Reviewer #2: Yes

6. Review Comments to the Author

Reviewer #2: The revised manuscript addresses all my comments raised in the review. However, there are two minor issues remaining:

1. Line 75 - Fasciola hepatica should be italicized

2. Line 156 - please provide the references to support this statement

7. PLOS authors have the option to publish the peer review history of their article (what does this mean?). If published, this will include your full peer review and any attached files.

Reviewer #2: No

---

## [Author Response · Author response to Decision Letter 2]

2 May 2026

We sincerely thank both reviewers for their valuable time, insightful comments, and constructive suggestions, which have significantly improved the quality of our manuscript.

Reviewers' comments

Reviewer 2

Comments 1: Line 75 - Fasciola hepatica should be italicized

• Response to comment: The scientific name F. hepatica has now been italicized as requested (Line 76).

Comments 2: Line 156 - please provide the references to support this statement

• Response to comment: The requested references have been added to support the statement

We hope that with these revisions, the manuscript now meets the standards for publication in PLOS ONE Journal.

---

## [Editor Report · Decision Letter 2]

5 May 2026

Fasciola hepatica Excretory-Secretory Products Attenuate Demyelination and Reduce Neuroinflammation in the Cuprizone –Induced Multiple Sclerosis Model

PONE-D-26-04172R2

Dear Dr. Shabani,

We’re pleased to inform you that your manuscript has been judged scientifically suitable for publication and will be formally accepted for publication once it meets all outstanding technical requirements.

Kind regards,

Tanja Grubić Kezele, Ph.D., M.D.

Academic Editor

PLOS One
---

## [Editor Report · Acceptance letter]

PONE-D-26-04172R2

PLOS One

Dear Dr. Shabani,

I'm pleased to inform you that your manuscript has been deemed suitable for publication in PLOS One. Congratulations! Your manuscript is now being handed over to our production team.

Kind regards,

on behalf of

Prof. dr. Tanja Grubić Kezele

Academic Editor

PLOS One